# Megafire affects stream sediment flux and dissolved organic matter reactivity, but land use dominates nutrient dynamics in semiarid watersheds

**Trevor Crandall**[1,2]*, **Erin Jones**[1], **Mitchell Greenhalgh**[1], **Rebecca J. Frei**[1,3],
**Natasha Griffin**[1], **Emilee Severe**[1], **Jordan Maxwell**[1], **Leika Patch**[1], **S. Isaac St. Clair**[4],
**Sam Bratsman**[1], **Marina Merritt**[5], **Adam J. Norris**[1], **Gregory T. Carling**[6], **Neil Hansen**[1],
**Samuel B. St. Clair**[1], **Benjamin W. Abbott**[1]

1 Department of Plant and Wildlife Sciences, Brigham Young University, Provo, Utah, United States of America, 2 Cimarron Valley Research Station, Oklahoma State University, Perkins, Oklahoma, United States of America, 3 Department of Renewable Resources, University of Alberta, Edmonton, Alberta, Canada, 4 Department of Statistics, Brigham Young University, Provo, Utah, United States of America, 5 Department of Chemical Engineering, Brigham Young University, Provo, Utah, United States of America, 6 Department of Geological Sciences, Brigham Young University, Provo, Utah, United States of America

* crandall.trevor@yahoo.com

**Data Availability Statement:** All relevant data are within the paper and its Supporting information files.

## Abstract

Climate change is causing larger wildfires and more extreme precipitation events in many regions. As these ecological disturbances increasingly coincide, they alter lateral fluxes of sediment, organic matter, and nutrients. Here, we report the stream chemistry response of watersheds in a semiarid region of Utah (USA) that were affected by a megafire followed by an extreme precipitation event in October 2018. We analyzed daily to hourly water samples at 10 stream locations from before the storm event until three weeks after its conclusion for suspended sediment, solute and nutrient concentrations, water isotopes, and dissolved organic matter concentration, optical properties, and reactivity. The megafire caused a ~2,000-fold increase in sediment flux and a ~6,000-fold increase in particulate carbon and nitrogen flux over the course of the storm. Unexpectedly, dissolved organic carbon (DOC) concentration was 2.1-fold higher in burned watersheds, despite the decreased organic matter from the fire. DOC from burned watersheds was 1.3-fold more biodegradable and 2.0-fold more photodegradable than in unburned watersheds based on 28-day dark and light incubations. Regardless of burn status, nutrient concentrations were higher in watersheds with greater urban and agricultural land use. Likewise, human land use had a greater effect than megafire on apparent hydrological residence time, with rapid stormwater signals in urban and agricultural areas but a gradual stormwater pulse in areas without direct human influence. These findings highlight how megafires and intense rainfall increase short-term particulate flux and alter organic matter concentration and characteristics. However, in contrast with previous research, which has largely focused on burned-unburned comparisons in pristine watersheds, we found that direct human influence exerted a primary control on nutrient status. Reducing anthropogenic nutrient sources

**Funding:** Ben Abbott; Sam St. Clair; Utah Division of Wildlife Resources; https://naturalresources. utah.gov/watershed-restoration-initiative#:~:text= Watershed%20Restoration%20Initiative%20%7C %20Utah%20Department%20of%20Natural% 20Resources&text=The%20Watershed% 20Restoration%20Initiative%20is,nearly%201.5% 20million%20acres%20statewide. The funders had no role in study design, data collection and analysis, decision to publish, or preparation of the manuscript.

**Competing interests:** The authors have declared that no competing interests exist.

could therefore increase socioecological resilience of surface water networks to changing wildfire regimes.

## Introduction

While ecosystem disturbance is crucial to the structure and function of the Earth's ecosystems [1–4], humans have accelerated the frequency and intensity of many disturbances, including wildfire and extreme precipitation [5–8]. Wildfires are becoming larger and more frequent in many parts of the world due to increased temperature, altered precipitation patterns, depleted snowpack, invasive plant species, and direct human ignition [5, 6, 9–13]. Megafires—defined as wildfires that exceed 400 $km^2$—have been rare historically, but they are increasing in frequency [14, 15], especially in the western US [16, 17]. Unlike smaller wildfires, which leave a mosaic of burned and unburned habitat types, megafires often affect whole regions creating large areas of homogeneous conditions [18]. Ecosystem succession following a megafire could differ fundamentally from smaller wildfires because multiple, adjacent ecosystem types are burned, increasing distances to unburned seed sources and intact habitat [19–21]. Megafires have serious consequences for human society as well, where they threaten human life and property, disrupt daily routines, impose economic costs from protecting or repairing infrastructure, increase insurance rates, and degrade air and water quality [22–29]. Because of their size and severity, megafires could change the magnitude and direction of interactions among wildlife habitat, watershed hydrology, and human management [30–34]. For example, small wildfires temporarily increase river runoff and nutrient loads [35–37], but a megafire affecting a whole river network could alter regional groundwater recharge and base-flow for many years [4, 38, 39], potentially increasing nutrient pressure on downstream ecosystems or otherwise altering food webs [40–43].

At the same time that wildfires are growing in size and severity throughout the U.S., precipitation is becoming more intense due to anthropogenic climate change and land use, with longer periods of water scarcity followed by intense storm events [6, 7, 44–46]. Together, these changes in disturbance regime threaten ecosystem function and human well-being by creating synchronous disturbances such as drought, flood, wildfire, and pollutant flux that affect large areas, potentially degrading habitat rather than restoring it [47–50]. Understanding how ecosystems in different biomes and human contexts respond to multiple stressors such as wildfire and extreme precipitation is crucial to support ecosystem integrity and services in the face of novel disturbance regimes [51–54].

Wildfire can influence downstream habitat and water available for human use by changing water routing through the watershed, which influences the amount and chemistry of surface and subsurface water [29, 55–57]. The hydrological effects of wildfire are complex and depend on multiple physical and biological attributes, including watershed size, aspect, vegetation type, energy balance, and snow versus rain dominance [58–61]. After a wildfire, the decrease in transpiration and increase in soil hydrophobicity can augment peak flows during storm events, which increases the frequency of flooding [41, 56, 62]. At small scales (e.g., catchments smaller than 100 $km^2$), this generally results in higher runoff ratios and lower groundwater recharge during precipitation events [52, 63], though observed responses range widely depending on local conditions [4, 50, 64]. At larger scales (e.g. >1,000 $km^2$), wildfire tends to cause moderate increases (e.g. 5 to 10%) in annual discharge and larger increases (up to 20%) during the summer low-flow period because of greater groundwater recharge from hydrologically losing river reaches [4, 50, 56].

The hydrological changes described above interact with the biological and physical changes caused by wildfires to affect solute and particulate chemistry in soil, surface water, and groundwater [42, 57, 65]. The loss of vegetation and soil organic matter during a wildfire can destabilize soils, leading to increased erosion rates that typically last around 5 years [56, 66, 67]. While sediment flux in rivers and lakes is commonly viewed as a negative phenomenon, this sediment deposition can restore banks, bars, and other fluvial features degraded by artificial flow control [39, 40]. However, increased sediment flux can also incur substantial costs to water treatment and distribution utilities [67–69]. In addition to sediment loss, the decrease in nutrient demand from terrestrial vegetation [19, 70, 71] and the deposition of nutrient-rich ash [2, 72] can substantially increase lateral nutrient flux for long periods after a wildfire (e.g. >10 years) depending on the recovery timeframes of terrestrial and aquatic vegetation communities [63, 73, 74]. These changes in hydrochemical conditions can trigger a shift in aquatic food webs, favoring algal productivity within the stream network and consequently contribute to altering aquatic nutrient retention [2, 75–79].

In this context, we investigated concentrations and fluxes of sediment, nutrients, and organic matter during an intense precipitation event after a megafire in the western U.S. Four overarching questions motivated our study: 1. How do megafires and extreme precipitation events affect nutrient loading to downstream water bodies, 2. What forms of nutrients are mobilized during these events (e.g. particulate versus dissolved, and organic versus inorganic), 3. How are the photodegradability and biodegradability of dissolved organic matter (DOM) affected by megafires, and 4. How do sediment, carbon, and nutrient fluxes from burned watersheds compare with watersheds impacted by direct human land use (e.g. urbanization and agriculture)? While we expected the extreme precipitation event to be a hot moment of material transport for all of the streams in our study [48, 80–82], we hypothesized that the type and amount of material transported during the storm would depend on the interaction of area burned and the direct human footprint. Specifically, we hypothesized that increased nutrient availability and decreased nutrient demand would result in higher lateral nutrient fluxes in burned watersheds compared to unburned watersheds [19, 70], and that point and nonpoint nutrient sources would result in higher lateral nutrient fluxes in human-influenced watersheds compared to natural watersheds. We also hypothesized that the loss of terrestrial plant matter and the creation of pyrogenic compounds in the soil would result in decreased photo- and biodegradability of DOM in burned watersheds [83–85]. Finally, we hypothesized that areas with a substantial direct human footprint would show higher variability in water chemistry parameters during the storm compared to more natural areas—whether burned or unburned— because of changes in hydrological routing [48, 86, 87]. To test these hypotheses, we collected daily to hourly water samples at 10 stream locations from immediately before an extreme precipitation event until three weeks after its conclusion. We analyzed these samples for nutrients, DOM, solute chemistry, and water isotopes, which we interpreted in a multi-tracer framework [53, 55, 88].

## Methods

### The Utah Lake watershed

All study sites are within the Utah Lake watershed (Fig 1), which is a part of the Great Salt Lake Watershed in the northeastern Great Basin. This high-elevation, semiarid region is endorheic (not draining to the ocean), and rivers flow to Utah Lake and then the Great Salt Lake. The Utah Lake watershed has a rapidly growing population, totaling ~500,000 in 2010 and is expected to surpass one million by 2060 [89]. This area is experiencing rapid climate change, particularly in the summertime with increasing temperatures, decreasing snowpack,

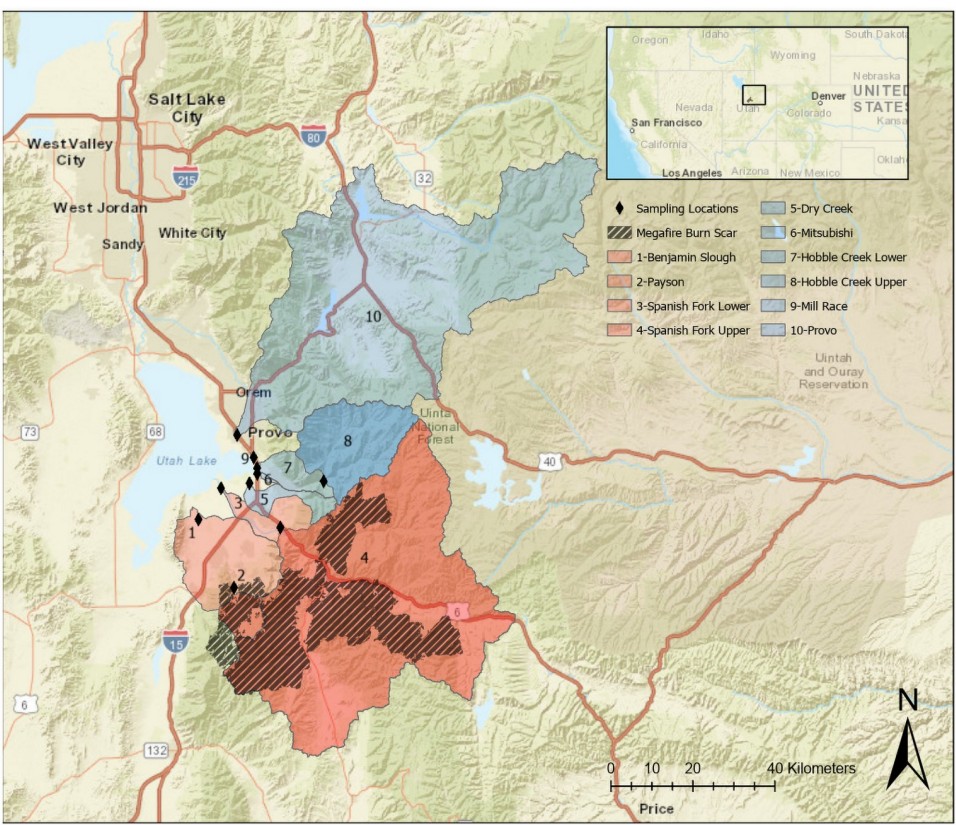

**Fig 1. Map showing the megafire burn scar, sampling locations, and contributing watersheds.** Watersheds are displayed as either red (burned) or blue (unburned) with numbers indicating the river or stream name in the legend. The map was created using opensource data from the USGS National Map and the Earth Resources Observation and Science Center.

and altered precipitation patterns [90–92]. Together, these pressures have resulted in extensive water diversions and inter-basin transfers, including in the studied watersheds [89], where water from the Colorado River basin is diverted into the Spanish Fork River (Fig 1). Geography in the Utah Lake watershed is diverse, ranging from its highest point on Mt. Nebo (3,636 m) to its lowest point at Utah Lake (1,368 m). Vegetation types covary with elevation, transitioning from a sagebrush steppe community in the valley bottoms to pinion-juniper forests, oak-maple forests, aspen-conifer forests, and alpine tundra at the highest elevations [93–95]. The area receives an average of 432 mm of rain and 117 mm of snow water equivalent per year. Mean daily high and low temperatures are respectively 33°C and 15°C in July and 2.2°C and -6.7°C in January.

Utah Lake is one of the largest natural lakes in western North America with a surface area of 384 km$^2$ and a contributing watershed area of 9,960 km$^2$ [89]. Utah Lake is shallow (mean depth of 2.74 m), and hypereutrophic, experiencing frequent cyanobacterial blooms because of a combination of natural risk factors (e.g. phosphorus-rich geology) and extensive anthropogenic nutrient loading [95–99]. Two tributaries contribute over half of the inflow into Utah Lake: the Spanish Fork River where the megafire occurred (24% of flow), and the Provo River which was unaffected by the megafire (30% of flow). Almost half of the water that flows into Utah Lake returns to the atmosphere via evaporation, with the remainder of the water moving

as surface or subsurface flow into the Jordan River valley northward toward the Great Salt Lake [100, 101].

## Megafire and hurricane characteristics

Extremely low snowpack, record precipitation, and high average summer temperatures set the scene for the three large wildfires in the Utah Lake watershed that converged into a megafire in 2018. The 128 km$^2$ Coal Hollow fire was started on August 4$^{th}$ by a lightning strike. The Bald Mountain and Pole Creek wildfires were also started by lightning on August 24$^{th}$ and September 6$^{th}$, burning 75 km$^2$ and 413 km$^2$ respectively, abutting the Coal Hollow burn scar. Together, these wildfires burned a contiguous area of 616 km$^2$, creating a megafire referred to as the Pole Creek Fire Complex (Fig 1). Aircraft-based thermal imaging during the wildfires and satellite based spectral imaging before, during, and after the fires (utahfireinfo.gov) classified over a third of the burned area (216 km$^2$) as high or extreme burn severity (>50% organic matter combustion and substantial alteration of soil structure), with the remaining two-thirds classified as moderate or low severity [102]. The relatively high proportion of high or extreme severity fire is attributable to the dry fuel, high temperature, and extreme wind behavior at the time of the megafire.

In October of 2018, as the megafire was still smoldering, the remnants of the Pacific storm Hurricane Rosa delivered intense rain to the burn scars. Hurricane Rosa was a category 4 hurricane in the eastern Pacific that made landfall in Baja California after being downgraded to a tropical storm. Precipitation from Hurricane Rosa affected Mexico and the southwest US as far north as Wyoming. Utah was hit by several waves of intense precipitation from October 3$^{rd}$-10$^{th}$. Throughout the week, approximately 105 mm of precipitation fell on the burn scars, more than 25% of the annual average precipitation for this area. Based on an analysis of precipitation observations from 2006 to 2021 (National Centers for Environmental Information), this level of weekly precipitation occurs approximately every 3 years in this area, representing a rare but increasingly common event [103]. Indeed, given the observed and predicted increases in extreme precipitation for this region and globally [6, 7, 44–46], this series of storms provides a useful analogue for potential future conditions.

## Experimental design and study sites

In early October, when we learned that Hurricane Rosa was likely to affect the megafire burn scar, we began collecting samples at 10 stream locations in the Utah Lake watershed (Fig 1 and S1 Fig). These locations included a variety of watershed sizes, burn coverages (0 to 90%), ecosystem types, and land uses (Table 1). Four sampling locations had burned areas in their watersheds, while six locations had no direct megafire influence. For one burned and one unburned watershed (sites 3 and 4 on the Spanish Fork River and sites 7 and 8 on Hobble Creek), we collected nested samples (i.e. longitudinally connected) that were upstream and downstream of major human infrastructure in the valley bottom (Fig 1 and S1 Fig). This allowed us to more directly quantify the influence of urban and agricultural development on water flow and chemistry. We determined catchment characteristics using the 2016 National Land Cover Database (www.mrlc.gov) [104]. We classified sampling sites as "human-influenced" or "natural" based on the sum of urban and agricultural land use in their watersheds (>10% and <10%, respectively). Even sites classified as natural had some grazing and recreational activities in their watersheds (Table 1). We aggregated vegetation classes into *forest* (deciduous, evergreen, and mixed forest generally over 5 m tall) and *herbaceous* (graminoid or herbaceous vegetation) categories.

**Table 1. Watershed characteristics of sampling sites within the Utah Lake watershed.**

| Site number and name | Area (km²) | Burned (%) | Mean slope (°) | Urban (%) | Impervious surface (%) | Agriculture (%) | Forest (%) | Herbaceous (%) |
|---|---|---|---|---|---|---|---|---|
| 1-Benjamin Slough | 319 | 67 | 21 | 10.6 | 3.7 | 18.0 | 38.4 | 6.4 |
| 2-Payson* | 54 | 90 | 29 | 2.68 | 0.0 | 0.0 | 82.9 | 0.5 |
| 3-Spanish Fork Lower | 1712 | 24 | 32 | 1.69 | 0.4 | 8.0 | 56.3 | 5.5 |
| 4-Spanish Fork Upper* | 1650 | 25 | 32 | 0.87 | 0.1 | 3.0 | 57.5 | 5.3 |
| 5-Dry Creek | 30 | 0 | 2 | 54.5 | 23.8 | 15.0 | 2.98 | 8.6 |
| 6-Mitsubishi Race | 3 | 0 | 1 | 61.2 | 28.0 | 0.1 | 2.81 | 1.8 |
| 7-Hobble Creek Lower | 324 | 0 | 41 | 4.48 | 1.7 | 1.5 | 56.7 | 7.6 |
| 8-Hobble Creek Upper* | 280 | 0 | 42 | 0.51 | 0.0 | 1.2 | 60 | 7.2 |
| 9-Mill Race | 47 | 0 | 36 | 40.6 | 23.0 | 0.5 | 47.1 | 2.9 |
| 10-Provo River | 1769 | 0 | 27 | 5.19 | 0.9 | 7.0 | 65.1 | 3.5 |

*Sites categorized as "natural" (less than 10% urban and agricultural land use)

## Sampling methods

To assess the short-term impacts of megafire and extreme precipitation, we used manual and automatic samplings throughout the month of October. Daily samples were taken from October 2nd-10th, every other day from the 11th-18th, and then periodically throughout the rest of October. The samples were collected in pre-rinsed high-density polyurethane bottles (triple rinsed with deionized water). At each site, we filtered 60 mL of water through a 0.45 μm filter (polyethersulfone membrane) into one bottle and collected a 1L bottle of unfiltered sample for particulate analysis. Samples were placed in a cooler with ice in the field until they were returned to the laboratory, where they were refrigerated until analysis.

On two of the burned rivers (sites 2 and 4), we deployed auto samplers (Teledyne ISCO) to collect hourly samples. The sampler at site 2 was washed away in a debris flow during the initial storm surge. The remaining auto sampler collected hourly samples into pre-rinsed HDPE bottles for the first four days, then every two hours until the sampler was retrieved on the ninth of October. We retrieved samples daily and transported them on ice to the laboratory where they were vacuum filtered through 47 mm diameter glass fiber filters (0.7 μm-effective pore size). Most samples were filtered within a few days of collection, but because of the high volume of samples and the extreme turbidity in the burned watersheds, some samples remained refrigerated but unfiltered for more than a week. Before filtration, we agitated samples vigorously to resuspend sediment. We determined suspended sediment concentration gravimetrically after drying filters at 105°C for 24 hours. We quantified organic carbon and total nitrogen in sediment by elemental analysis (Vario EL Cube, Vienna Austria) after acidification to remove inorganic carbon.

Two sampling locations were near USGS water discharge stations (sites 4 and 7), allowing us to calculate areal sediment flux. As our sampling was relatively high-frequency, we used linear interpolation between samplings to estimate sediment concentration, which we multiplied by the 15-minute discharge data from the USGS stations.

## Solutes and isotopes

To test our hypotheses about the effects of megafire and human disturbance on water chemistry, we analyzed filtered samples for a wide range of water chemistry parameters. We analyzed samples on an inter-coupled plasma spectrometer (Thermo Scientific 4700 series) for dissolved elements, including trace metals (Al, As, B, Ba, Ca, Cd, Co, Cr, Cu, Fe, K, Mg, Mn, Mo, Na, Ni,

P, Pb, S, Se, Si, Sr, Ti, V and Zn). We quantified major anion and cation concentrations using ion chromatography (Dionex, Thermo Fisher Scientific), including inorganic nutrients and a variety of ions (Fluoride, Acetate, Formate, Chloride, Nitrite, Bromide, Nitrate, Sulfate, Phosphate, Lithium, Sodium, Ammonium, Potassium, and Magnesium).

To test how megafire and land use affected carbon and nutrient availability, we quantified several fractions of carbon, nitrogen, and phosphorus. We measured dissolved organic and inorganic carbon (DOC and DIC, respectively) with an elemental analyzer (Vario TOC Cube, Vienna Austria). We calculated dissolved inorganic nitrogen (DIN) by summing $NO_3^-$, $NO_2^-$, and $NH_4^+$. We calculated dissolved organic nitrogen (DON) by subtracting DIN from total dissolved nitrogen (Vario TOC Cube, Vienna Austria). We calculated dissolved organic phosphorus (DOP) as the difference between phosphate ($PO_4^{3-}$) and total dissolved phosphorus.

To test our hypothesis about DOM photo- and biodegradability following wildfire, we carried out a laboratory incubation (described in the following section) and used fluorescence spectroscopy (AQUALOG, Horiba) to quantify the optical properties of the DOM [55, 105, 106]. We analyzed the absorbance data and the excitation emission matrices (EEMs) to calculate several common indices of DOM composition [107–110], including biological index (BIX), humification index (HIX), fluorescence index (FI), spectral ratio (SR), and specific UV absorbance at 254 nm ($SUVA_{254}$) [108]. All samples were corrected for inner filter effects, Rayleigh scatter, and blank subtraction in MATLAB™ (version 6.9; MathWorks, Natick, Massachusetts), and samples that exceeded 0.3 absorbance units at excitation 254 nm were diluted with deionized water and re-run [111, 112]. We correlated these optical properties with DOM bio- and photodegradability to assess the importance of DOM composition in determining reactivity [55, 112, 113].

To assess how the megafire and land use affected water flow path and residence time, we analyzed water isotopes via cavity ringdown spectroscopy (Model DLT-100, Los Gatos Research, San Jose, CA, USA). Because Hurricane Rosa travelled rapidly from the ocean to Utah, it likely had a relatively enriched isotopic signature compared to the pre-event water, which was subjected to continental moisture recycling [114–116]. However, because we did not have precipitation samples, we could only calculate qualitative differences, rather than fractions of young and old water [55, 117]. During and after the storm event, we interpreted the relative changes in $\delta D$ and $\delta^{18}O$ of water to assess arrival of event water at the sampling site.

## DOM photodegradation and biodegradation

To test our hypothesis about how megafire and land use affect DOM photo- and biodegradability, we conducted a 28-day laboratory incubation. The ecological importance of DOM depends on its reactivity to sunlight and biological attack [112, 118, 119]. Reactive DOM can be rapidly mineralized into inorganic nutrients (e.g. DIC, $PO_4^{3-}$, and DIN), while nonreactive DOM may remain relatively inert in the ecosystem for years or centuries [120, 121]. We used water filtered through glass fiber filters (0.7 μm-effective pore size) for the incubation, following the standard protocol proposed by Vonk et al. [122]. The relatively coarse pore size allowed some of the endogenous microbial community to pass through the filter into the incubation bottles. To measure the photoreactivity of the DOM, we incubated half of the replicates in the light to quantify the importance of photomineralization (abiotic breakdown of DOM to $CO_2$ by light) and photostimulation (transformation of DOM into more biodegradable compounds) [123].

With water from each site, we poured 12 mL into three vials. The first vial ($t_0$) was immediately acidified to a pH < 2 with 5N HCl to preserve the sample until measurement on the

elemental analyzer. The other two vials were incubated for 28 days at 20˚C; one in the light (L) and one covered in tinfoil to create darkness (D). The intensity of the broad-spectrum radiation was 284 μmol m$^{-2}$s$^{-1}$ throughout the experiment to mimic typical light characteristics in stream water columns [124]. After the 28-day period, we acidified the L and D vials to stop microbial activity and remove DIC before analysis for DOC on the elemental analyzer. We calculated the concentration and percentage of the DOC that disappeared during the experiment, which we interpreted as the biodegradable DOC (BDOC) and photodegradable DOC (PDOC) for the D and L treatments, respectively. We calculated photopriming as the difference in BDOC and PDOC, with a positive photopriming value representing more DOC loss in the light treatment. We note that our experimental design, which depended on DOC disappearance, does not allow us to distinguish photomineralization from photostimulation, meaning that our estimates of PDOC may include both phenomena [118, 119, 125].

### Statistical analyses

We used both parametric and nonparametric statistical tests and visualizations to evaluate our hypotheses. To test for differences among categories (human vs. natural and burned vs. unburned), we used analysis of variance (ANOVA) with Tukey HSD for multiple comparisons. We tested assumptions visually (e.g. normal distribution, homoscedasticity) with residual and quantile-quantile plots, and used natural log transformations when needed. Most of the distributions were symmetrical, which was surprising given how skewed water chemistry data can be [126, 127]. To account for spatial dependence of samples collected at the same location (i.e. repeated measures), we included site as a blocking factor. We correlated carbon and nutrient concentrations with chloride (Cl$^-$)—a common indicator of urban and agricultural activity [55]—to assess how much nutrients were naturally derived or anthropogenic for the various land use classes. To test for differences in BDOC and PDOC (i.e. for the presence of photopriming), we used paired T-tests. We used simple and multiple linear regression to quantify relationships between optical properties of the DOM quantified with the scanning fluorometer and the BDOC and PDOC values quantified in the incubation experiment [53]. To assess if the assumption of linearity was infringed, we visually inspected biplots and compared coefficients with nonparametric rank correlations. For the multiple regressions, we standardized all predictors before analysis to allow comparison of coefficients as a metric of influence, and we visually examined collinearity, homogeneity of residuals, and linearity, following standard methods [19]. All analyses and visualizations were performed in the R statistical computing software [128].

## Results

### Effects of wildfire and land use on sediment concentration and flux

For the October 2$^{nd}$-10$^{th}$ period, the areal sediment yield was 1974-fold higher at the burned site than the unburned site: 5.9 kg/km$^2$ of sediment at the unburned, human-influenced sampling location where we had discharge data (7-Hobble Creek Lower) versus 11,651 kg/km$^2$ at the burned, natural sampling location where we had discharge data (4-Spanish Fork Upper; Fig 2B). The delivery of ash, sediment, and other material to Utah Lake created a plume that was visible from space as it spread throughout October depending on lake currents (Fig 2C). For both human-influenced and natural sampling locations, sites affected by the megafire had more total suspended sediment (TSS) than sampling locations not directly affected by the megafire (F$_{1,241}$ = 8.5, p = 0.004; Fig 2). Unburned sampling locations generally had less than 1 g/L of TSS throughout the observation period, while burned sampling locations ranged from less than 1 g/L to 80 g/L (Fig 2A). A large pulse of TSS was transported during and directly

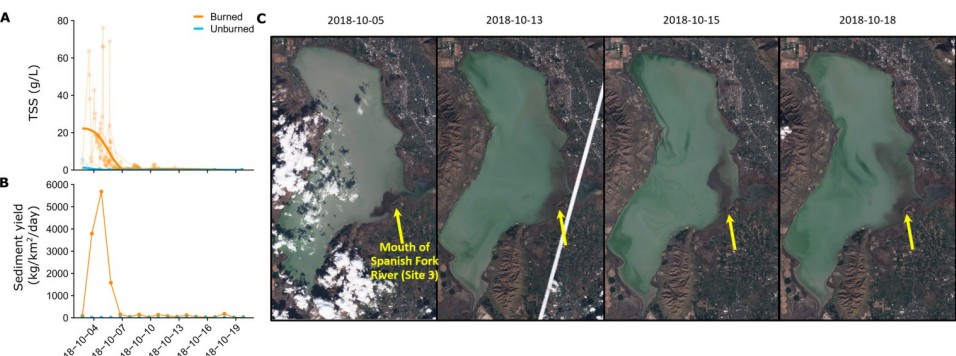

**Fig 2. Sediment concentration and flux from burned and unburned watersheds.** Total Suspended Sediment (TSS) is the mass of particles retained during filtration to 0.7 μm expressed per liter. Sediment yield is the flux of sediment normalized by watershed area. **A)** Individual measurements for each site (translucent points and lines) and the smoothed mean for burned and unburned sites (solid blue and orange lines). **B)** Cumulative daily values of sediment yield for the burned and unburned site with a USGS gauging station. **C)** Satellite imagery from the storm period. Copernicus Sentinel data retrieved from ASF DAAC in 2019, processed by ESA.

following the storm event from October 2 to 7, but TSS concentrations decreased rapidly in the following days (Fig 2). There was not a significant difference in TSS concentration between human-influenced and natural sampling locations ($F_{1,241}$ = 1.44, p = 0.23). Neither particulate organic C nor particulate N showed statistically significant differences with land use ($F_{2,169}$ = 2.37, p = 0.126). However, the particulate organic C content was approximately 3-fold higher in burned sites than in unburned sites (2.9% ± 1.1 versus 0.93% ± 0.46; mean ± SD; $F_{1,62}$ = 42.6, p = 1.65×10$^{-8}$), as was the particulate nitrogen (0.21% ± 0.11 versus 0.069% ± 0.032; $F_{1,109}$ = 25.78, p = 1.59×10$^{-6}$ S2 Fig). Particulate C:N ratio was 13.5 for both burned and unburned watersheds. Together with the elevated sediment flux, this meant that particulate C and N flux from burned watersheds was approximately 6,100-fold higher than in unburned watersheds.

## Hydrological response and solute chemistry

We interpreted the stable isotopes of water as metrics of flow path and residence time. During the storm period, $\delta^{18}$O was higher in human-influenced areas ($F_{1,212}$ = 11.5, p = 0.00082; Fig 3A), indicating dominance of rapid flow paths that efficiently transferred storm water to the river network. Human-influenced sampling locations showed a more rapid increase in $\delta^{18}$O during the onset of the storm and more temporal variability throughout the sampling period compared to natural locations. For natural sampling locations, unburned sites had lower $\delta^{18}$O, suggesting a higher proportion of pre-event water and higher catchment-level residence times (Fig 3). Deuterium (D‰) showed significant differences by burn status ($F_{1,212}$ = 8.769, p = 0.00341) and land use ($F_{1,212}$ = 22.274, p = 4.29×10$^{-6}$; Fig 3B). As expected, patterns for D ‰ were similar to $\delta^{18}$O, with more rapid increases in human-influenced watersheds, slower increases in burned but natural locations, and slowest increases in unburned and natural locations (S3 Fig).

Carbon, nitrogen, and phosphorus species showed a variety of differences based on human influence and burn status. DOC was only significantly higher at burned natural sites compared to unburned natural sites ($F_{1,170}$ = 5.63, p = 0.019; Fig 3D). DIC was significantly higher at burned sites for both human-influenced and natural watersheds ($F_{1,170}$ = 4.75, p = 0.03). DIN

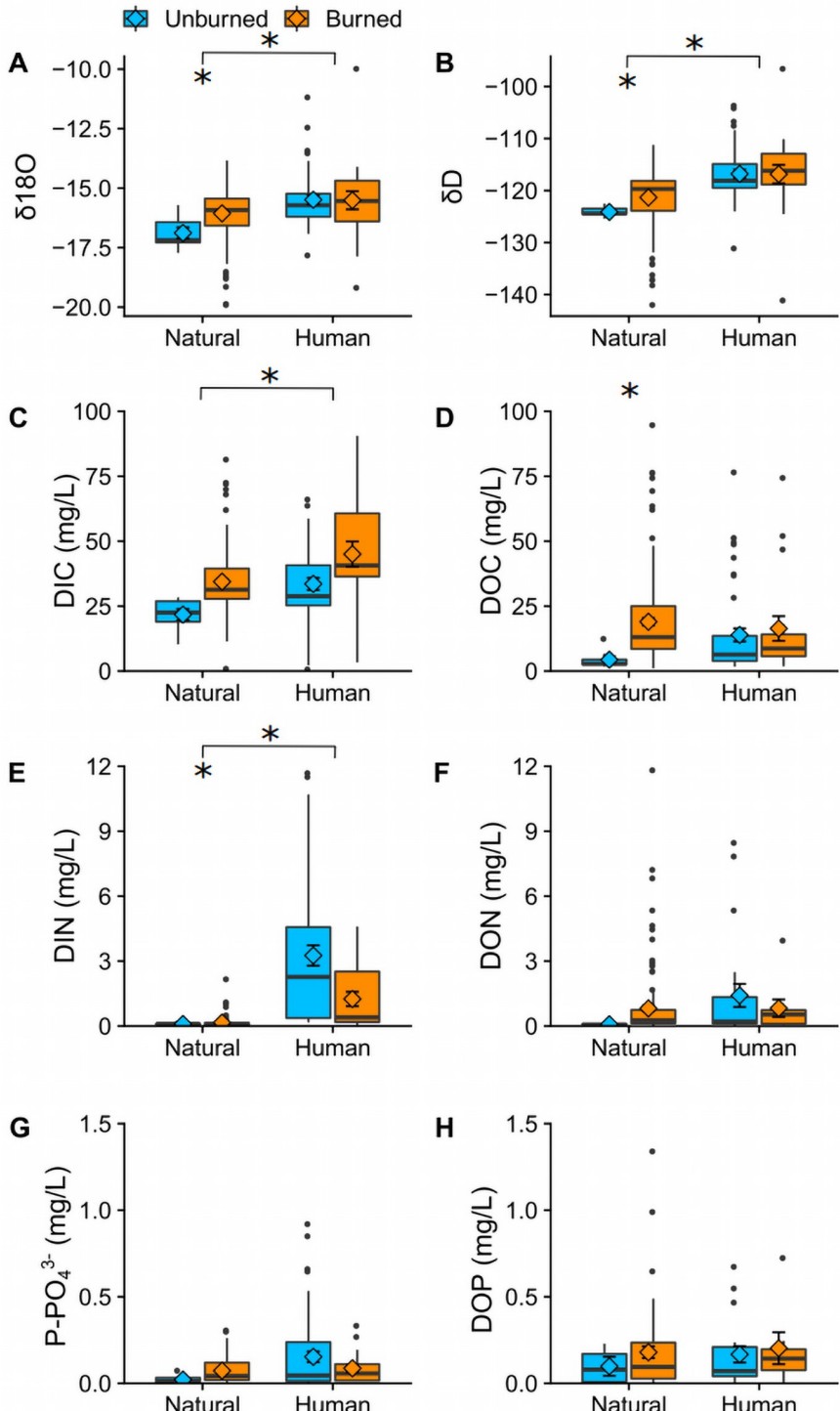

**Fig 3. Water isotopes and carbon, nitrogen, and phosphorus species during the observation period categorized by burn status and land use.** Statistically significant differences denoted by an *. A-B) Water isotopes, C-D) Dissolved inorganic and organic carbon, E-F) Dissolved inorganic and organic nitrogen, G-H) Phosphate and dissolved organic phosphorus.

differed by both land use and burn status ($F_{1, 236}$ = 8.782 p = 0.003), with much higher DIN in human-influenced watersheds, and somewhat higher DIN at burned sites in natural watersheds. DOP and $PO_4^{3-}$ did not show significant differences by burned status or land use (Fig 3).

There were strong positive correlations between Cl⁻ (a common indicator of human activity) and most nutrients at urban and agricultural sites ($R^2$ > 0.6, p < 0.001), but a very weak relationship at natural sites ($R^2$ < 0.1, p < 0.05; Fig 4).

## Dissolved organic matter optical properties and degradability

The optical properties of DOM differed by burn status and land use (Fig 5), indicating substantial shifts in DOM source and level of processing. Natural, burned sampling locations had a much higher Fluorescence Index (FI; $F_{1, 111}$ = 14.6, p = 0.0002) compared to natural unburned locations or human-influenced locations, suggesting more microbially-derived DOM relative to plant-derived DOM. Biological Index (BIX) was significantly higher at burned locations ($F_{1, 111}$ = 46.95, p = $4.27 \times 10^{-10}$) regardless of whether the sample was from a human-influenced or natural watershed, potentially indicating recently produced DOM from aquatic sources. Spectral Ratio (SR) was lower at burned, natural sampling locations ($F_{1, 111}$ = 11.46, p = 0.00099), indicating higher molecular weight compounds. Absorbance at 254 nm (ABS) was highly variable, but highest at unburned, human-influenced sampling locations, indicating greater aromatic DOM (Fig 5D).

The 28-day incubation experiment showed that burned watersheds had more photoreactive DOC, with approximately twice the percentage of PDOC (14% ± 6, SD) compared to unburned watersheds (7% ± 3; $F_{1, 91}$ = 46, p = $1.16 \times 10^{-9}$; Fig 6). Because DOC concentration was also generally higher at burned locations, the differences in total DOC consumed were even larger (Fig 6A). BDOC in mg/L was higher in burned watersheds for both natural and human-influenced areas ($F_{1,95}$ = 8.57, p = 0.004; Fig 6B). Regardless of land use, burned sampling locations had higher percent PDOC (14% ± 5%) than unburned sampling locations (8% ± 6; p adjusted = 0.008; Fig 6C). Likewise, BDOC % was higher in burned ($F_{1,95}$ = 8.57, p = 0.004), though it did not differ with land use ($F_{1,95}$ = 1.24, p = 0.26; Fig 6D).

The magnitude and sign of photopriming differed with burn status (Fig 6E). For the burned sites, nearly all the samples experienced positive photopriming, consistent across land uses, while the unburned sites showed negative photopriming, or little to no photopriming, indicating fundamental differences in DOM structure and reactivity ($F_{2,105}$ = 3.7, p = 0.28; Fig 6E). The photopriming results followed the same general pattern as the concentration: smaller differences at human locations, but extremely marked differences in the natural locations (Fig 6E).

The relationships between DOM optical properties and reactivity differed for BDOC and PDOC (Fig 7 and S3 Fig). FI and BIX were positively correlated with PDOC (R > 0.3, p < 0.005), while SR was negatively correlated with PDOC (R = -0.26, p = 0.025; S3 Fig). BDOC was only correlated with ABS (R = -0.41, p = 0.0002), showing a moderate, negative relationship (S3 Fig). When combined in the multiple linear regression with nutrient parameters, only DIN and ABS were significant predictors of BDOC, though the model only predicted 26% of the variation in BDOC (Fig 7). FI, DIN, ABS, and HIX were all significant predictors of PDOC, with the model explaining 47% of the variation in PDOC.

To assess how much inorganic nutrient could be released from degrading DOM, we multiplied the PDOC percentages by the DON and DOP concentrations (Fig 8). The degradable organic nitrogen and phosphorus was relatively small compared to the elevated inorganic nutrients, especially at sites in human-influenced watersheds.

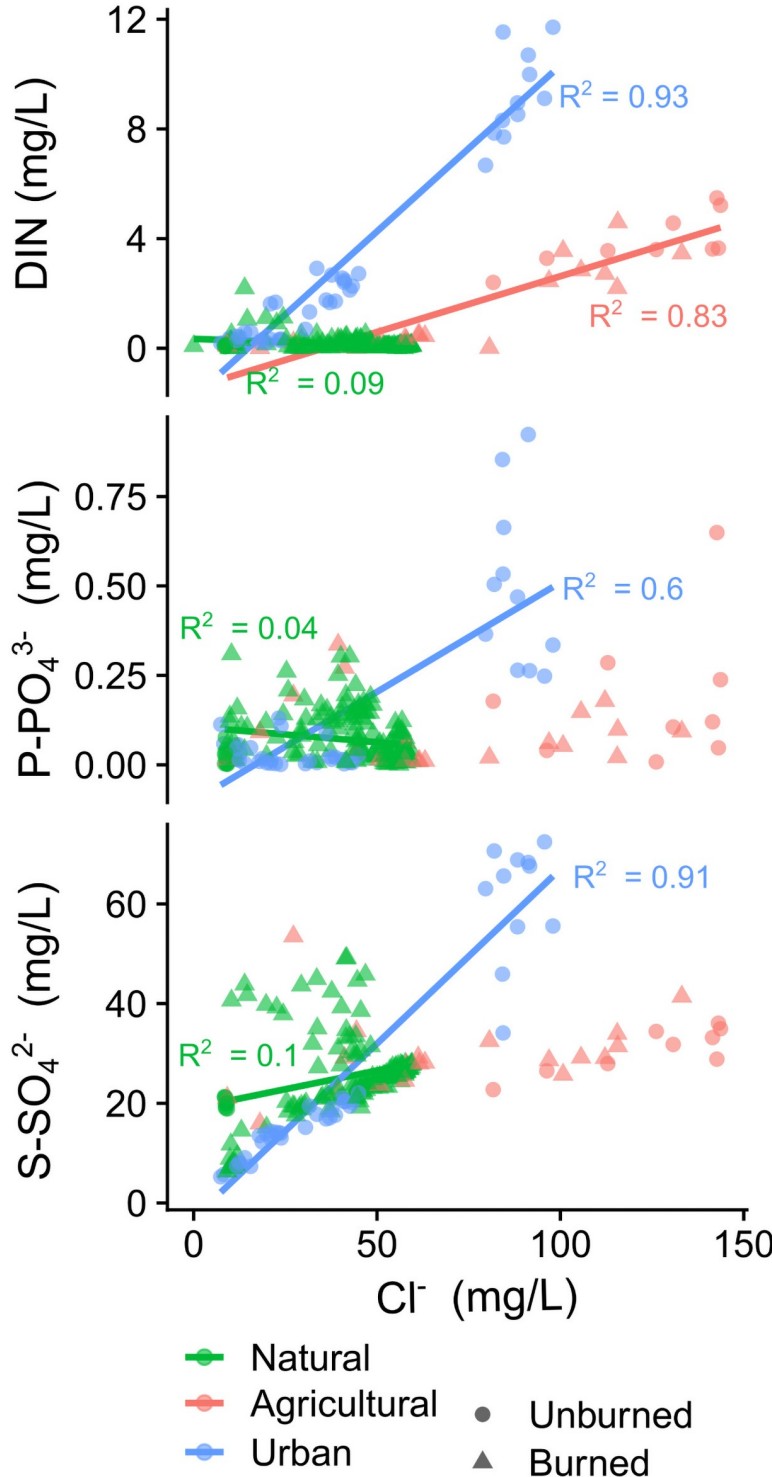

**Fig 4. The relationship between chloride (Cl⁻), a common indicator of human activity and inorganic nutrients in natural, agricultural, and urban watersheds.** Circles and triangles represent unburned and burned watersheds, respectively.

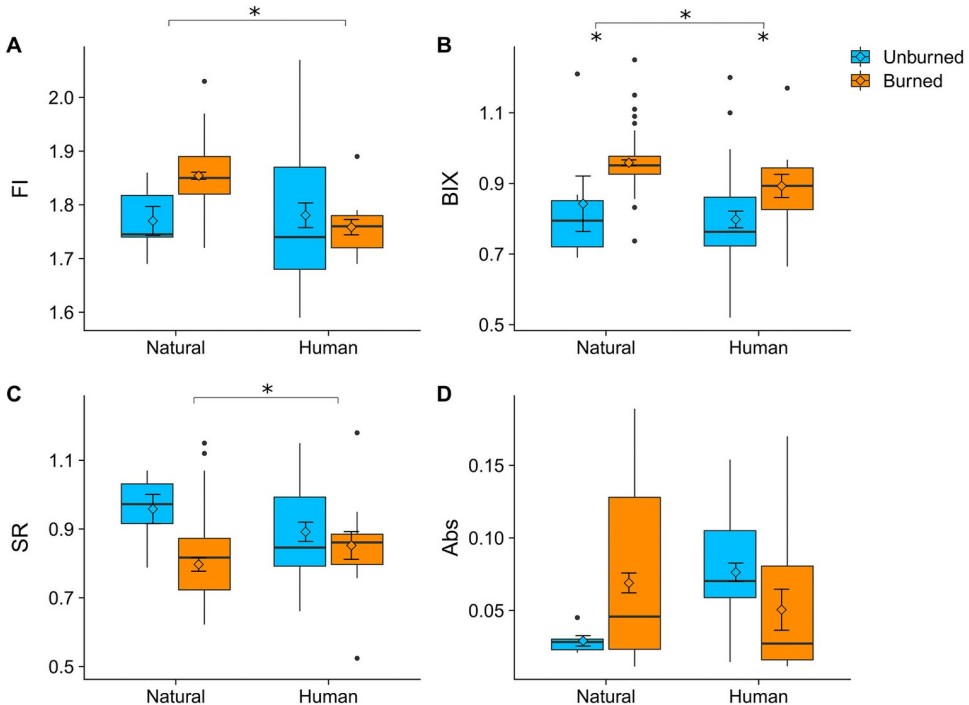

**Fig 5. Optical properties of dissolved organic matter (DOM) from burned and unburned sites in natural and human watersheds.** Optical parameters included fluorescence index (FI), biological index (BIX), slope ratio (SR), and absorbance (Abs). See [129] for detailed interpretation of optical parameters.

## Discussion

In this study, we measured how megafire and human land use alter lateral transport of sediment and nutrients in semiarid watersheds. We hypothesized that the megafire and direct human footprint would interact to determine the effects on hydrochemistry, with higher nutrient concentrations and less bioavailable organic matter in burned watersheds. We found that while the megafire strongly influenced sediment dynamics and DOM properties, the direct human footprint more strongly influenced inorganic nutrient concentrations. Contrary to our hypothesis, burned watersheds had substantially higher DOM reactivity in both light and dark incubations, indicating that this megafire either created reactive pyrogenic compounds or reduced less reactive DOM sources, resulting in a net increase of DOM photo and biodegradability.

### How does megafire affect lateral material flux?

The size and severity of the Pole Creek Megafire in combination with the intense storm event caused enormous amounts of sediment transport from the terrestrial environment, in line with previous studies [67, 130–133]. Even though our estimates do not include bedload, if the material was evenly distributed across the lakebed, the amount of material transported to Utah Lake during the week after the storm amounted to 49,900 kg/km². Utah Lake is experiencing frequent cyanobacterial blooms, partially due to nutrient release from the lakebed [89, 96, 98], and understanding the fate of this material is central to predicting its impact on lake nutrient state, and the general effects of megafire on downstream eutrophication risk [133, 134]. Because megafire sediment contains organic and inorganic nutrients, it could increase nutrient availability in receiving waterbodies, exacerbating eutrophication [66, 98, 133, 135]. On the

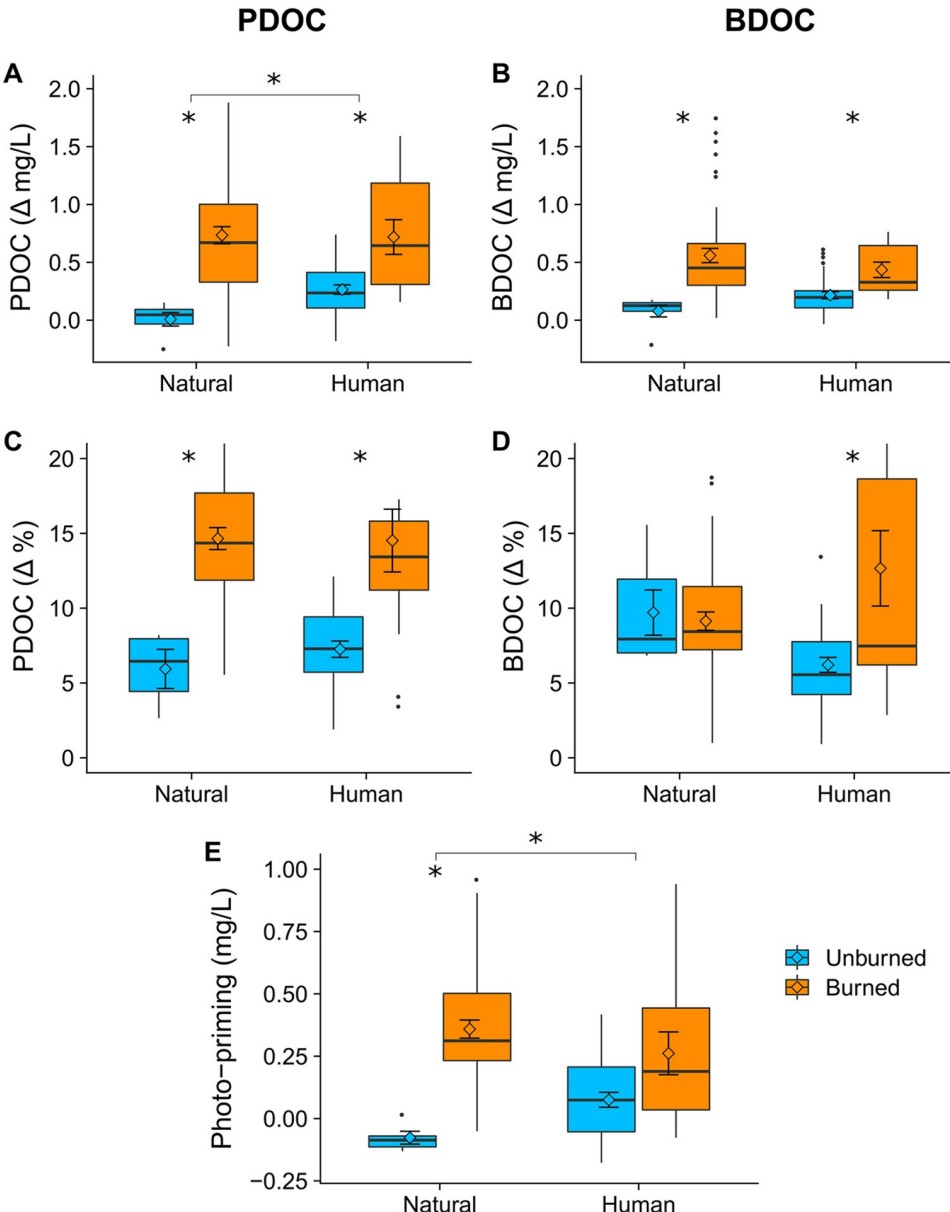

**Fig 6. Changes in dissolved organic carbon (DOC) concentration and percent during a 28-day incubation experiment of samples collected at stream sites based on burn and land use status.** Photodegradable DOC (PDOC) is the concentration and percent change for samples incubated under broad-spectrum radiation (see Methods) while biodegradable DOC (BDOC) is from the dark incubation. Panels A and B show the concentration of DOC (mg/L) that was lost during the incubation experiment. Panels C and D show the percentage of DOC lost during the experiment. Panel E shows the pairwise differences between A and B, representing the additional amount of DOC consumed due to exposure to light (i.e. photopriming).

other hand, the particulate N load from the wildfire and storm event remains small compared to nutrient inputs from wastewater plants, urban runoff, agricultural return flows, and atmospheric deposition [98, 100, 136].

While our study only documents the release of sediment immediately after a wildfire, prior research suggests enhanced sediment flux from severe wildfire can last 2 to 10 years or longer [3, 66, 67, 137, 138]. This enhanced erosion can cause costly disruptions of water treatment

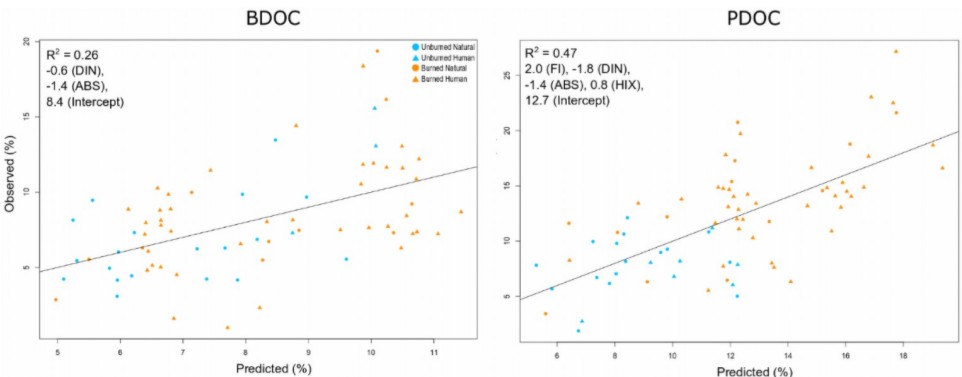

**Fig 7. Predicted versus observed BDOC and PDOC based on the multiple linear regression with optical properties of DOM and background nutrients.** Burn status and land use class are indicated by the color and shape of each point.

[29], but the ecological effects of burn-induced sediment on complex downstream ecosystems remain difficult to predict [138–140]. For example, increased sediment loading can limit light availability in affected water bodies, potentially decreasing photosynthesis and eutrophication temporarily [98, 141]. Likewise, the delivery of charred but bioavailable organic matter with a high carbon to nutrient stoichiometry [142, 143] could result in at least temporary nutrient immobilization, potentially attenuating the impact of fire-mobilized material or even providing a short-term nutrient sink. New data and processing methods have made available wildfire scar data and remotely sensed algal bloom data on continental scales [133, 144, 145]. Spatio-temporal analysis of these and other data are needed to further improve our understanding of the effects of wildfire on eutrophication in reservoirs, lakes, and estuaries.

While megafire had a clear and dominant influence on sediment dynamics, its effects on carbon and nutrients were more complex. Following observations from similar megafires [42, 65, 146], we hypothesized that increased nutrient availability and decreased nutrient demand in the terrestrial environment would result in higher lateral nutrient fluxes in burned watersheds. However, in contrast with previous research, which primarily focuses on burned-unburned comparisons in pristine watersheds [42, 72, 73, 147], we found that the presence of urban and agricultural activity exerted a much greater influence on nutrient status than the megafire. We explore these unexpected results in the following sections.

## Wildfire versus land use effects on nutrient dynamics

While we did observe nutrient increases in the natural, burned watersheds compared to natural, unburned watersheds, in line with previous research [73, 146], the largest nutrient effects in our study were in areas with a direct human footprint. The influence of human land use on water and nutrient dynamics has been observed around the world [44, 148–151]. Agriculture and urbanization strongly influence nutrient concentrations in groundwater and surface water [152–154]. However, in the context of wildfire, this research is often overlooked, with wildfire studies focusing on comparisons of pristine (i.e. minimal direct human influence) burned and unburned watersheds [39, 67, 73, 147].

Investigating pristine watersheds is helpful to isolate burn effects, but in the context of the Anthropocene [44, 155], understanding the consequences of wildfire in relation to other disturbances is needed. Our study confirms that megafire can trigger nutrient release from pristine watersheds [146], but that other sources of excess nutrients and pollutants—in this case wastewater effluent, urban stormwater, and agricultural runoff—have a larger effect on overall

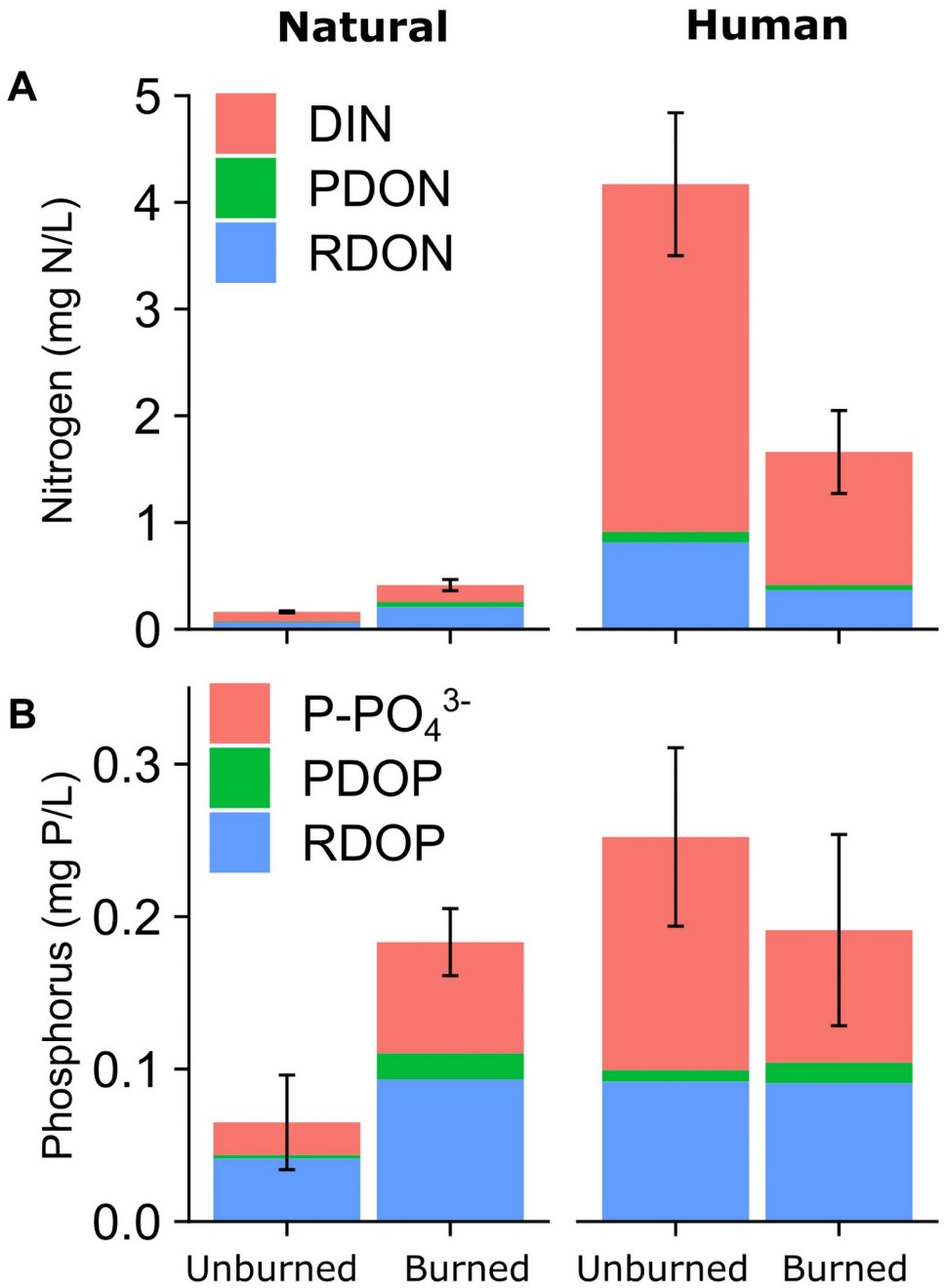

**Fig 8. The overall concentration (mean ± SE) and composition of nitrogen (N) and phosphorus (P) in burned and unburned watersheds for natural and human land use classes.** The photodegradable and recalcitrant fractions were calculated by multiplying the PDOC percent measured in the incubations with the concentration of DON and DOP.

inorganic nutrient concentrations and fluxes [156–158]. A recent study of water chemistry throughout the Utah Lake watershed confirms this direct human domination of nutrient sources and flux [95]. Together, these observations suggest that direct human impact, not megafire, is the primary threat to aquatic ecosystems in semiarid ecosystems [141, 159, 160]. In this light, we recommend reducing nutrient fluxes from urban and agricultural areas to make ecosystems more resilient to megafire and extreme precipitation events [53, 161, 162].

## Why is burned DOM so degradable?

Contrary to our hypotheses, we observed higher photo and biodegradability at burned sampling locations. We expected that the loss of terrestrial plant matter [163–165] and the creation of high-temperature pyrogenic compounds during the megafire [83, 85, 142] would result in less reactive compounds in burned watersheds. However, pyrogenic organic matter varies widely in composition and reactivity [28, 166–168]. The effects of combustion on organic matter properties depend on initial substrate (e.g. plant or soil sources), the characteristics of the combustion process (e.g. temperature, duration, and percentage consumption), and the degree of processing during subsequent transport to and through the hydrological network [82–85, 142, 169]. In general, low temperature combustion (< 200˚C) creates more degradable compounds than when combustion occurs at higher temperatures [84, 169, 170]. Even though a third of this megafire had high or extreme severity, two-thirds of the burn scar experienced more moderate combustion conditions. Additionally, because more organic matter is consumed in severely burned areas, the moderate or low severity areas likely contain a larger portion of the postfire organic matter. Consequently, postfire DOM could derive primarily from areas with moderate or low severity burn, which are likely to create more reactive and soluble DOM [85, 166, 170].

Additionally, or alternatively, the substantial increases in BDOC and PDOC could indicate that the source and composition of DOM were changed in non-additive ways that do not scale from laboratory experiments to the watershed level. For example, we observed a decrease in residence time (i.e. faster arrival of the stormflow water), which could be associated with fresher and more photo and biodegradable DOM [120, 170–172]. The consistently positive photo-priming we observed could indicate that photodegradation is an important pathway for DOM transformation and degradation, in contrast with recent studies, which have found that photomineralization and photopriming are largely limited to low-light environments [118, 119, 173]. In low-light environments such as the Arctic, photodegradation rates have been found to exceed biodegradation rates in the dark by a factor of nearly five [118], which is higher, though in the same order of magnitude, of our observations where PDOC was roughly double BDOC. Testing how general this pattern of photopriming following wildfire will be important to better predicting the persistence and ecological consequences of megafire on downstream water quality.

One explanation for the increased PDOC is that sediment shading is protecting pyrogenic DOM from photodegradation during transport. Removing the sediment prior to the optical property experiment could effectively expose DOM to the light for the first time, resulting in rapid photomineralization and photostimulation [125, 170]. A second explanation for the high PDOC and BDOC could be new production of DOM via algal growth (i.e. autochthonous DOM). Increased nutrients and sunlight following megafires from removal of riparian vegetation can stimulate in-stream production of DOM and particulate matter [75, 77, 78, 174]. However, given that the megafire had just occurred before the storm event, and the fact that extreme sediment loads likely precluded any autochthonous production, this in-stream PDOC production hypothesis seems unlikely. A third factor could be that increased nutrient availability in the burned sites primed the biological or photochemical breakdown of the DOM. Higher nutrient availability can accelerate DOM mineralization via several stoichiometric and thermodynamic phenomena [112, 120, 175]. The addition of nutrients has also been recently observed to increase the breakdown of photo-sensitive colored DOM, which acts as an important protection from UV radiation in many aquatic and marine ecosystems [112]. If these findings are general, megafire and the associated release of sediment and

altered DOM could have substantial and potentially costly consequences for downstream ecosystems and communities.

## Implications for megafire management in the Anthropocene

The increasing intensity and frequency of wildfire in the western US and other regions threatens ecosystems and communities already under substantial stresses [176–178]. While this study suggests that downstream nutrient dynamics are more influenced by human land use than wildfire, it confirms the strong influence of wildfire and extreme precipitation on sediment and DOM delivery. Because wildfires in many regions occur upstream of municipal water sources, the risk of sediment blockage and DOM contamination of drinking water sources remains serious [34, 179, 180]. Likewise, the increasing extent and severity of wildfires in many regions threaten human life and infrastructure at the wildland-urban interface and human health at continental scales via air pollution [24, 181–185].

We propose to prioritize education and awareness of the causes of wildfire and the capacity and limits of local management to effectively respond to rapidly changing wildfire regimes. Public knowledge and buy-in are central to implementation of the most effective methods for increasing resilience to wildfire in the Anthropocene: 1. Eliminating greenhouse gas emissions to stop climate change, 2. Limiting development of the wildland-urban interface, and 3. Using prescribed burns to reduce extreme wildfire risk [13, 185–187].

Concerning the causes of change in wildfire regime, parts of the western US remain in a state of fire deficit, where past and present fire suppression has limited fire at unsustainably low levels [188]. Simultaneously, anthropogenic climate change has caused long-term increases in fire extent, especially severe wildfire, associated with dry fuels, extreme vapor pressure deficits, and extended fire seasons [3, 5, 8, 189]. While mitigating climate change by reducing fossil fuel emissions is the only tenable solution to the latter problem, climate and megafires themselves make improving management more difficult in two connected ways. First, climate change has reduced the safe windows for prescribed burning, one of the most effective local management tools [8, 13, 17, 190]. Second, destructive and violent megafire events cause local trauma, potentially contributing to unwillingness to allow prescribed burns [191, 192]. For example, the 2018 megafire in this study resulted in the evacuation of more than 10,000 residents and severe local air pollution during the burning. Though this megafire was likely associated with both a historical wildfire deficit and climate change, its intensity and violence made the public less willing to tolerate prescribed burns in the area in the following years. In 2019 and 2020, several moderate wildfires were suppressed in the Utah Lake watershed because of persistent public opposition, though these fires would have achieved management goals safely and economically. Ultimately, to address the growing wildfire crisis in the western US, we need to rehabilitate the public image of natural and prescribed wildfire, while also making clear that the current surge in megafire extent and severity is associated with anthropogenic climate change.

## Supporting information

**S1 Fig. Detailed maps of the study watersheds, burn scar severity, topography, and land use.** We created the maps with ArcGIS Pro (ESRI) using open source basemap layers from the global GIS user community (the USGS National Map and Earth Resources Observation and Science Center).
(DOCX)

**S2 Fig. Particulate organic carbon and particulate nitrogen content (%) based on elemental analysis of material that did not pass through a glass fiber filter with an effective pore size of 0.7 μm.** There was insufficient sample size to The boxplots represent the median and its 95% confidence interval (the notches), the interquartile range (IQR), and points beyond 1.5 times the IQR.
(DOCX)

**S3 Fig. Individual pairwise comparisons of optical properties and biodegradable and photodegradable dissolved organic carbon (BDOC and PDOC).** Linear fit lines and 95% confidence intervals shown when p < 0.05.
(DOCX)

**S4 Fig. Time series of deuterium values for all the sites individually and the smoothed means for agricultural (green), urban (red), and natural (blue) sites.**
(DOCX)

**S1 Data.**
(XLSX)

# Acknowledgments

We thank Leslie Lange, Rachel Watts, Michelle Baker, Rhetta Shoemaker, and Zach Aanderud for assistance in the field and laboratory. We recognize Shanae Tate for compiling the satellite imagery shown in Fig 2. We thank the U.S. Forest Service and Utah County for their assistance and guidance in site selection and access.

# Author Contributions

**Conceptualization:** Trevor Crandall, Erin Jones, Gregory T. Carling, Neil Hansen.

**Data curation:** Trevor Crandall, Mitchell Greenhalgh, Rebecca J. Frei, Natasha Griffin, Emilee Severe, Jordan Maxwell, Leika Patch, S. Isaac St. Clair, Sam Bratsman, Marina Merritt, Gregory T. Carling, Neil Hansen, Samuel B. St. Clair.

**Formal analysis:** Trevor Crandall, Erin Jones, Rebecca J. Frei.

**Funding acquisition:** Benjamin W. Abbott.

**Investigation:** Trevor Crandall, Rebecca J. Frei, Benjamin W. Abbott.

**Methodology:** Trevor Crandall, Gregory T. Carling, Neil Hansen.

**Project administration:** Trevor Crandall.

**Resources:** Trevor Crandall.

**Software:** Trevor Crandall.

**Supervision:** Trevor Crandall, Benjamin W. Abbott.

**Visualization:** Trevor Crandall, Mitchell Greenhalgh, Benjamin W. Abbott.

**Writing – original draft:** Trevor Crandall.

**Writing – review & editing:** Trevor Crandall, Erin Jones, Mitchell Greenhalgh, Natasha Griffin, Emilee Severe, Jordan Maxwell, Leika Patch, S. Isaac St. Clair, Sam Bratsman, Marina Merritt, Adam J. Norris, Gregory T. Carling, Neil Hansen, Samuel B. St. Clair, Benjamin W. Abbott.

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
