## [Decision Letter · Decision Letter 0]

22 Apr 2021

PONE-D-21-06624

Megafire affects stream sediment flux and dissolved organic matter reactivity, but land use dominates nutrient dynamics

PLOS ONE

Dear Dr. Crandall,

Thank you for submitting your manuscript to PLOS ONE. After careful consideration, we feel that it has merit but does not fully meet PLOS ONE’s publication criteria as it currently stands. Therefore, we invite you to submit a revised version of the manuscript that addresses the points raised during the review process.

Both reviewers agreed the manuscript is interesting and well-written but raised relevant questions that need to be clarified before the paper is accepted for publication. More specifically, they recommend put your study into a broader context, highlighting how the unique storm you analyzed could be analogous to other post-fire rain events; add details about the specific impacts of the land-use classes on nutrient use and export and the spatial scale; focus the discussion on what it is directly relevant to the study findings.

We look forward to receiving your revised manuscript.

Kind regards,

Angelina Martínez-Yrízar, Ph.D.

Academic Editor

PLOS ONE

Journal Requirements:

In your Methods section, please provide additional location information of the sampling sites, including geographic coordinates for the data set if available.

In your Methods section, please provide additional information regarding the permits you obtained for the work. Please ensure you have included the full name of the authority that approved the field site access and, if no permits were required, a brief statement explaining why.

We note that Figures 1, 2 and S1 in your submission contain map images which may be copyrighted. All PLOS content is published under the Creative Commons Attribution License (CC BY 4.0), which means that the manuscript, images, and Supporting Information files will be freely available online, and any third party is permitted to access, download, copy, distribute, and use these materials in any way, even commercially, with proper attribution. For these reasons, we cannot publish previously copyrighted maps or satellite images created using proprietary data, such as Google software (Google Maps, Street View, and Earth). For more information, see our copyright guidelines: http://journals.plos.org/plosone/s/licenses-and-copyright.

4a, You may seek permission from the original copyright holder of Figures 1, 2 and S1 to publish the content specifically under the CC BY 4.0 license. 

4b, If you are unable to obtain permission from the original copyright holder to publish these figures under the CC BY 4.0 license or if the copyright holder’s requirements are incompatible with the CC BY 4.0 license, please either i) remove the figure or ii) supply a replacement figure that complies with the CC BY 4.0 license. Please check copyright information on all replacement figures and update the figure caption with source information. If applicable, please specify in the figure caption text when a figure is similar but not identical to the original image and is therefore for illustrative purposes only.

Please include captions for your Supporting Information files at the end of your manuscript, and update any in-text citations to match accordingly. Please see our Supporting Information guidelines for more information: http://journals.plos.org/plosone/s/supporting-information.

Reviewers' comments:

Reviewer's Responses to Questions

**Comments to the Author**

1. Is the manuscript technically sound, and do the data support the conclusions?

Reviewer #1: Yes

Reviewer #2: Partly

2. Has the statistical analysis been performed appropriately and rigorously? 

Reviewer #1: No

Reviewer #2: No

3. Have the authors made all data underlying the findings in their manuscript fully available?

Reviewer #1: Yes

Reviewer #2: Yes

4. Is the manuscript presented in an intelligible fashion and written in standard English?

Reviewer #1: Yes

Reviewer #2: Yes

5. Review Comments to the Author

Reviewer #1: This is a well written manuscript about an interesting combination of wildfire followed by an unusual rainstorm. In that sense, it is not widely applicable to the audience because the circumstances are so unique. I think the authors could make the paper more applicable to a wider audience by highlighting how this unique storm could be analogous to other post-fire rain events. They will need to add more literature on this and could analyze the storm itself better in comparison to typical rain events in the region. I think it is a good manuscript with some interesting findings. I have several recommendations for the authors to improve the manuscript. This includes re-evaluating the statistical approach they used and how they highlight (or don't) highlight their findings with statistics. They make interesting points in the results and discussion but do not support them with statistics, just general inferences. I also would like more information on the study site locations relative to one another and why they subsetted them into burned and unburned and human and natural, seems that subsetting 10 sites makes the comparisons weaker between 2 sites here and 2 sites there. These are all things the authors could and should address. The approach and methods are interesting, and the discussion is good. There are a few nice take away points for a reader. I recommend the author fix a few things and re-submit.

My more detailed recommendations and edits are below:

Review of: Megafire affects stream sediment flux and dissolved organic matter reactivity, but land use dominates nutrient dynamics

Author: Crandall, Trevor

PONE D-21-06624

Line 64: isn’t this degradation temporary? This sentence makes it sound permanent, but there is evidence of recovery from the literature

Lines 92-98 not sure this paragraph is necessary. Unless you plan to discuss macroinvertebrate communities, this paragraph is about one of the many possible ecosystem changes that can happen as a result of fire but are unrelated to sediment transport. You are overlooking that the algal growth is a result of a change in the canopy cover, not necessarily a nutrient loading response.

Line 131: remove the word “very”

Line 155: signs of high or extreme burn severity? Are you saying this qualitatively? Have you analyzed the burn scar’s burn severity? If you are to include burn severity at all, it should not be qualitative. There are remote sensing methods to quantify this. Otherwise do not include burn severity in the introduction. A visual assessment of a portion of the fire is not enough.

Line 168: Please add more detail about the 10 locations, which are on the same stream? This would help me (your reader) understand the study design a little better before I refer to the table or figures. Your map is not easy to decipher streams and could be improved

Line 215: change “dissolve” to “dissolved”

Lines 229 – 236: After reading this paragraph I went back to see how this method fit into your research questions. How does the isotope analysis help answer any of your research questions? You might need to set this up better in the introduction to further support why you did this. Otherwise it seems like an extra method that you did, kind of threw the kitchen sink at this period of sampling, without any guidance for why or what you were expecting to see. So introduce it better in the introduction or explain it in this paragraph, otherwise it seems extra and not related to your research focus

Line 263: pretty unusual to use parametric statistics on stream water quality data. Refer to Helsel and Hirsch et al. 2020 Statistical Methods in Water Resources

Line 280: I am getting confused by the term “human sampling locations” You are using the 10% development in land use threshold as a way of sub-setting your 10 sites into two categories. Along with a comparison of burned to unburned sites. I am not sure this subsetting adds to your story. Also, doesn’t subsetting your 10 sites to human and natural reduce your sample size so that you are really comparing 2 -3 sites to 2 -3 other sites? It may be more appropriate and clearer to highlight human land use as a single component to compare results as one paragraph in Results section and one in Discussion.

I like how for the remaining Results section you compared sites- simply burned and unburned, it was easier to follow and made for a cleaner comparison.

Line 407: I don’t love this sentence. “effects of the megafire were dominant on sediment dynamics” instead of “dominant on” which I find to be a strange choice could you change to “determined” or “controlled”?

Lines 422-426: This is a great finding and discussion point. This to me is one of your biggest take-away messages.

Linr 465: typo- change “sconed” to “second”

Line 465-471: I think you can remove the allochthonous and autochthonous section from the introduction and leave your explanation in this discussion paragraph, you introduce it and wrap it up nicely here.

Lines 480-496: This feels like partly introduction material re-visited and opinion. Of course I agree with your opinion, I am not sure it fits here with your detailed water quality study. I would recommend removing it

Results & Discussion-

You mention using ANOVA, T-Test and regression but yet there is really no mention of statistical results in your results section or your discussion. Why? You highlight differences, but were they statistically significant?

I am disappointed to not see more storm analysis. Your study was based on one huge storm. I think that makes this situation unique and not necessarily applicable to all regions. Is there a way you can look at the volume of precipitation received in that storm relative to other post-fire storm events in the literature? Do you think that if a smaller rainstorm had occurred right after the Pole Complex fire the sediment and nutrient impact would be the same? I guess I am asking you to highlight your unique storm situation as you did in the introduction in the results and discussion as well. I also recommend highlighting how while it was a uniquely huge storm, it is analogous to other rainstorm driven sediment events after fire.

Figures: all the figures are a bit fuzzy

Reviewer #2: This is an interesting and well-written article. Placing wildfire effects into a larger context of watershed change and comparing them to other sources of water quality perturbation is worthwhile. For example, the main message that urban and agricultural areas can have larger effects on N and P is not surprising, but the comparison with wildfire effects is unique. The comparison of C quality is also unique and worthwhile. I’ve included several post-fire C composition papers for reference.

The water quality responses to various land use changes scale with their extent and severity, so the challenge this paper faces is to develop ‘fair’ or useful comparisons between wildfire, urbanization and agriculture. The sample areas span 4 orders of magnitude in size, sample areas are nested and most contain multiple land cover types. There is need to add details about the specific impacts of the land use classes on nutrient use and export and about the spatial scale of the various study areas. Quite simply, where did the nutrients from the agricultural and urban areas come from? Is this fertilized, irrigated row-crop agricultural, suburban lawns, pastures, industry,…? Without such information, readers are unable to interpret these findings.

Regarding fire, past studies indicate that wildfire effects scale with the proportion of a contributing area that is severely burned and thus some fires have little appreciable effect on water quality. It is not clear to what extent the catchments that are included in this study were impacted by high severity wildfire. There is a statement that > 1/3 of the area burned hot (lines 154-155). It’s unclear how that estimate corresponds to the specific study sites identified in Table 1. Please replot the map with watershed boundaries and burn severity. As mentioned above, provide additional detail regarding the forest, urban and agricultural areas.

The combined effects of the wildfire and storm are interesting, but it’s difficult to know how to interpret the findings from this combined disturbance. Would there have been no water quality effects without a severe storm? It seems like that the patterns would have been different for a less severe storm event. What is the threshold, for example of post-storm nutrient losses from the various agriculture or agricultural areas? If this was an unprecedented storm it is not surprising that the nutrient losses would exceed the conditions of stormwater collection and urban and agricultural water quality best management practices. Quite simply, the authors need to clarify if these findings are a unique response to the combined fire and storm and if so this qualification should be presented and discussed from the outset. The paper currently presents the findings as response to wildfire across land use/land cover types rather than a combined fire x storm event.

19 Specify the dates of the fire and storm in the abstract so readers will know how long it had been between the fire, flood, sampling, etc.

Line 165 Table 1

What is meant by herbaceous? Where are range or shrublands included?

Specify the type of agriculture and especially whether this includes irrigated or fertilized row crops. Specify if the urban areas include suburban residential developments with irrigated and fertilized lawns.

Line 175 Is grazing classified with agriculture?

Line 207 Where are all these analytes reported?

Line 371 Should this be Fig 8?

Discussion

377-378 It’s unclear how the hypothesized interaction reflects the direct human footprint? It seems that the stated response (higher nutrients/less bioavailable OM) relates to burning, but not human activity? Please revise to clarify the link.

384-386 Not sure this sentence is needed as a segue.

394 past tense? ‘amounted to’

439-440 The loss of terrestrial plant matter would be a year/multi-year response. During this study you’re measuring the initial flush of charred, partially charred and unburned OM. This makes sense over a long-term, but in the immediate post-fire period, the export of

448-450 Numerous studies have reported increased DOC and reactive C following both wildfire and lab heating experiments.

463-464 What experiment?

465 “second factor” ?

470 Yes, and scour during the flood would also have removed any existing algae/biofilms, so in stream autrophs would not have had much effect during the period of this study.

471 Are you referring to increased nutrient transfer from burned soils/uplands?

477 The post-fire increase in nutrients will contribute to and augment these concerns. Include it to the list here.

479-510 The authors’ clear opinions make for interesting reading here.

Reference Format Please check and standardize.

Additional References to consider

Cawley, K.M., A.K. Hohner, D.C. Podgorski, W.T. Cooper, J.A. Korak, and F.L. Rosario-Ortiz. 2017. Molecular and Spectroscopic Characterization of Water Extractable Organic Matter from Thermally Altered Soils Reveal Insight into Disinfection Byproduct Precursors. Environmental Science & Technology 51:771-779.

Chow, A.T., K.-P. Tsai, T.S. Fegel, D.N. Pierson, and C.C. Rhoades. 2019. Lasting Effects of Wildfire on Disinfection By-Product Formation in Forest Catchments. Journal of Environmental Quality 48:1826-1834.

Hohner, A.K., K. Cawley, J. Oropeza, R.S. Summers, and F.L. Rosario-Ortiz. 2016. Drinking water treatment response following a Colorado wildfire. Water Research 105:187-198.

Hohner, A.K., C.C. Rhoades, P. Wilkerson, and F.L. Rosario-Ortiz. 2019. Wildfires Alter Forest Watersheds and Threaten Drinking Water Quality. Accounts of Chemical Research 52:1234-1244.

Writer, J.H., A. Hohner, J. Oropeza, A. Schmidt, K. Cawley, and F.L. Rosario-Ortiz. 2014. Water treatment implications after the High Park Wildfire in Colorado. J. Am. Water Works Assoc 106:85-86.

6. PLOS authors have the option to publish the peer review history of their article (what does this mean?). If published, this will include your full peer review and any attached files.

Reviewer #1: No

Reviewer #2: No

---

## [Author Response · Author response to Decision Letter 0]

1 Sep 2021

Dear Editor,

We are very grateful for the detailed and constructive feedback from you and the reviewers. We have now carefully revised our manuscript based on this input. We provide point-by-point responses below (reviewer comments in black, our responses in blue). Our apologies for the delay in the resubmission, which was due to multiple pandemic-related issues. We hope that this does not disqualify our manuscript from review.

During the revisions, we solicited substantial help from Adam Norris, who was previously acknowledged for his technical assistance and editing in the Acknowledgements section. Given his added contributions, we now include him as a co-author.

Please let us know if we can provide additional information and thank you for your consideration,

Trevor Crandall on behalf of all co-authors

We have formatted the manuscript to match all style requirements. 

2. In your Methods section, please provide additional location information of the sampling sites, including geographic coordinates for the data set if available.

We now provide latitude and longitude for the main sites. More detailed information (i.e., coordinates for all sites) is included in the supplementary data.

 We have updated the methods to include information on permissions and permits (e.g., in the acknowledgements). 

4. We note that Figures 1, 2 and S1 in your submission contain map images which may be copyrighted. All PLOS content is published under the Creative Commons Attribution License (CC BY 4.0), which means that the manuscript, images, and Supporting Information files will be freely available online, and any third party is permitted to access, download, copy, distribute, and use these materials in any way, even commercially, with proper attribution. For these reasons, we cannot publish previously copyrighted maps or satellite images created using proprietary data, such as Google software (Google Maps, Street View, and Earth). For more information, see our copyright guidelines: http://journals.plos.org/plosone/s/licenses-and-copyright.

We require you to either (1) present written permission from the copyright holder to publish these figures specifically under the CC BY 4.0 license, or (2) remove the figures from your submission… 

All the included maps are fully compatible with PLOS ONE’s CC BY 4.0 license. We updated the captions accordingly: We created the maps with ArcGIS Pro (ESRI) using open source basemap layers from the global GIS user community (the USGS National Map and Earth Resources Observation and Science Center).

1. Please include captions for your Supporting Information files at the end of your manuscript, and update any in-text citations to match accordingly. Please see our Supporting Information guidelines for more information: http://journals.plos.org/plosone/s/supporting-information.

Updated.

Reviewer #1: This is a well written manuscript about an interesting combination of wildfire followed by an unusual rainstorm. In that sense, it is not widely applicable to the audience because the circumstances are so unique. I think the authors could make the paper more applicable to a wider audience by highlighting how this unique storm could be analogous to other post-fire rain events. They will need to add more literature on this and could analyze the storm itself better in comparison to typical rain events in the region. I think it is a good manuscript with some interesting findings. I have several recommendations for the authors to improve the manuscript. This includes re-evaluating the statistical approach they used and how they highlight (or don't) highlight their findings with statistics. They make interesting points in the results and discussion but do not support them with statistics, just general inferences. I also would like more information on the study site locations relative to one another and why they subsetted them into burned and unburned and human and natural, seems that subsetting 10 sites makes the comparisons weaker between 2 sites here and 2 sites there. These are all things the authors could and should address. The approach and methods are interesting, and the discussion is good. There are a few nice take away points for a reader. I recommend the author fix a few things and re-submit.

We thank the reviewer for their thoughtful assessments and criticisms. We have revised the manuscript thoroughly taking these comments into consideration. Specifically, we better contextualize this storm event (commenting on how it is unique and typical of other events), and we clarified the experimental design and statistical approach, which are now better integrated throughout.

Here is some of the new text on the representativeness of the storm:

Throughout the week, approximately 105 mm of precipitation fell on the burn scars, more than 25% of the annual average precipitation for this area. Based on an analysis of precipitation observations from 2006 to 2021 (National Centers for Environmental Information), this level of weekly precipitation occurs approximately every 3 years in this area, representing a rare but increasingly common event [97]. Indeed, given the observed and predicted increases in extreme precipitation for this region and globally [6,7,38–40], this series of storms provides a useful analogue for potential future conditions.

My more detailed recommendations and edits are below:

Line 64: isn’t this degradation temporary? This sentence makes it sound permanent, but there is evidence of recovery from the literature- 

We agree with the reviewer, and throughout the manuscript, we are trying to make the point that wildfire usually has a positive effect on habitat. This sentence is talking about modified disturbance regimes, including interactions among wildfire and hydrological extremes. In these scenarios, there are many plausible consequences of wildfire that are harmful, either temporarily or permanently, including restructuring of the stream channel, facilitation of invasive species, and destruction of remaining habitat fragments. We have revised throughout to try to make this distinction clear: natural wildfire regime and hydrological variability are necessary and positive, but modified regimes may bring unexpected risks.

Lines 92-98 not sure this paragraph is necessary. Unless you plan to discuss macroinvertebrate communities, this paragraph is about one of the many possible ecosystem changes that can happen as a result of fire but are unrelated to sediment transport. You are overlooking that the algal growth is a result of a change in the canopy cover, not necessarily a nutrient loading response- 

This paragraph was originally included to get the reader thinking about food web consequences of disturbance, including issues of eutrophication and changing light availability. Nevertheless, we see the reviewer’s point about it being unnecessary, and we have shortened this section, integrating it with the previous paragraph. 

Line 131: remove the word “very”- 

Removed

Line 155: signs of high or extreme burn severity? Are you saying this qualitatively? Have you analyzed the burn scar’s burn severity? If you are to include burn severity at all, it should not be qualitative. There are remote sensing methods to quantify this. Otherwise do not include burn severity in the introduction. A visual assessment of a portion of the fire is not enough- 

Burn severity was analyzed quantitatively based on airborne thermal imaging during the wildfire and satellite analysis of pre- and post-fire spectral analysis. We did not collect these data ourselves, but we performed the analysis for the affected watersheds, which we now describe in the methods: 

Aircraft-based thermal imaging during the wildfires and satellite based spectral imaging before, during, and after the fires (utahfireinfo.gov) classified over a third of the burned area (216 km²) as high or extreme burn severity (>50% organic matter combustion and substantial alteration of soil structure), with the remaining two-thirds classified as moderate or low severity [96]. The relatively high proportion of high or extreme severity fire is attributable to the dry fuel, high temperature, and extreme wind behavior at the time of the megafire.

Line 168: Please add more detail about the 10 locations, which are on the same stream? This would help me (your reader) understand the study design a little better before I refer to the table or figures. Your map is not easy to decipher streams and could be improved-

We now list what sites are nested (i.e., occurring on the same stream network) in the table and text. We also have revised the methods section generally for clarity and flow.

Line 215: change “dissolve” to “dissolved”-

Changed. Thank you for catching this typo.

Lines 229 – 236: After reading this paragraph I went back to see how this method fit into your research questions. How does the isotope analysis help answer any of your research questions? You might need to set this up better in the introduction to further support why you did this. Otherwise it seems like an extra method that you did, kind of threw the kitchen sink at this period of sampling, without any guidance for why or what you were expecting to see. So introduce it better in the introduction or explain it in this paragraph, otherwise it seems extra and not related to your research focus- 

We have revised as suggested. Specifically, we more fully enumerate our hypotheses at the end of the introduction, and we justify the need for both hydrological and biogeochemical characterization in the introduction and methods.

Line 263: pretty unusual to use parametric statistics on stream water quality data. Refer to Helsel and Hirsch et al. 2020 Statistical Methods in Water Resources- 

We completely agree with the reviewer that selection of appropriate analytical tools is crucial to robust conclusions. Consequently, we relied on both parametric and nonparametric statistics. For example, when considering loads (or concentrations’ effects on loads), the parametric mean is the most meaningful parameter because nonparametric methods that discount extreme values would result in under or overestimates of material transported. Conversely, nonparametric evaluation of percentiles (as in the boxplots and confidence intervals of the medians) provides a more robust estimate of typical conditions experienced by in-stream organisms. Ultimately, we were surprised at the relatively symmetrical distributions of values, which resulted in acceptable concordance between parametric and nonparametric estimates. Likewise, though both linear and nonlinear methods were tested (e.g., product- and rank-based correlations), the linear methods proved adequate for these data. We now provide more justification for the selection of tests and satisfaction (or not) of assumptions.

Line 280: I am getting confused by the term “human sampling locations” You are using the 10% development in land use threshold as a way of sub-setting your 10 sites into two categories. Along with a comparison of burned to unburned sites. I am not sure this subsetting adds to your story. Also, doesn’t subsetting your 10 sites to human and natural reduce your sample size so that you are really comparing 2 -3 sites to 2 -3 other sites? It may be more appropriate and clearer to highlight human land use as a single component to compare results as one paragraph in Results section and one in Discussion.

I like how for the remaining Results section you compared sites- simply burned and unburned, it was easier to follow and made for a cleaner comparison- 

We recognize the difficulty of reading this non-standard terminology, and we have revised for clarity, now referring to these sites as “human-influenced”. We maintain the human-influenced versus pristine distinction, which is central to our study questions, but have tried to use more intuitive terms and natural language.

Line 407: I don’t love this sentence. “effects of the megafire were dominant on sediment dynamics” instead of “dominant on” which I find to be a strange choice could you change to “determined” or “controlled”?- 

We now used “determined”, as suggested by reviewer. 

Lines 422-426: This is a great finding and discussion point. This to me is one of your biggest take-away messages.

Thank you for the positive feedback. 

Linr 465: typo- change “sconed” to “second”-

Changed. Thank you for finding this embarrassing (and delicious) typo.

Line 465-471: I think you can remove the allochthonous and autochthonous section from the introduction and leave your explanation in this discussion paragraph, you introduce it and wrap it up nicely here.

Removed as suggested.

Lines 480-496: This feels like partly introduction material re-visited and opinion. Of course I agree with your opinion, I am not sure it fits here with your detailed water quality study. I would recommend removing it- 

We have revised this section to better relate it to our main study goals and findings. We would be happy to revisit or cut if the reviewer is still unsatisfied.

Results & Discussion-

You mention using ANOVA, T-Test and regression but yet there is really no mention of statistical results in your results section or your discussion. Why? You highlight differences, but were they statistically significant?- 

We apologize for the confusion. We tried to report statistical results throughout but accidentally omitted them in these last paragraphs. We typically do not report the statistical outputs in the discussion, which seeks to contextualize the findings more generally, but we do in the figures and results.

I am disappointed to not see more storm analysis. Your study was based on one huge storm. I think that makes this situation unique and not necessarily applicable to all regions. Is there a way you can look at the volume of precipitation received in that storm relative to other post-fire storm events in the literature? Do you think that if a smaller rainstorm had occurred right after the Pole Complex fire the sediment and nutrient impact would be the same? I guess I am asking you to highlight your unique storm situation as you did in the introduction in the results and discussion as well. I also recommend highlighting how while it was a uniquely huge storm, it is analogous to other rainstorm driven sediment events after fire.- 

In response to this important critique, we now provide more discussion about what aspects of the study are applicable elsewhere and what aspects are unique. For example, while the confluence of extreme events and wildfire has historically been rarer, with the artificial expansion of the wildfire season because of climate change and human ignition sources, wildfires are occurring more often during periods prone to extreme precipitation. Therefore, this specific event may be less representative of the historical fire conditions but quite representative of novel fire-hydrology interactions in the future.

Figures: all the figures are a bit fuzzy

We apologize for this technical difficulty. We uploaded high-resolution files as requested, but there may have been some compression in the conversion to PDF by the journal. We will make sure to resolve this if it persists.

Reviewer #2: This is an interesting and well-written article. Placing wildfire effects into a larger context of watershed change and comparing them to other sources of water quality perturbation is worthwhile. For example, the main message that urban and agricultural areas can have larger effects on N and P is not surprising, but the comparison with wildfire effects is unique. The comparison of C quality is also unique and worthwhile. I’ve included several post-fire C composition papers for reference.

We are very grateful for the reviewer’s careful read and positive assessment of our article. We have added the suggested references.

The water quality responses to various land use changes scale with their extent and severity, so the challenge this paper faces is to develop ‘fair’ or useful comparisons between wildfire, urbanization and agriculture. The sample areas span 4 orders of magnitude in size, sample areas are nested and most contain multiple land cover types. There is need to add details about the specific impacts of the land use classes on nutrient use and export and about the spatial scale of the various study areas. Quite simply, where did the nutrients from the agricultural and urban areas come from? Is this fertilized, irrigated row-crop agricultural, suburban lawns, pastures, industry,…? Without such information, readers are unable to interpret these findings.

We appreciate these important critiques. We now provide greater information on the land cover types, including what is known about nutrient sources in this watershed. While our study does not have an experimental design suited to establish relationships between the various types of land use and water chemistry (e.g. a small sample size relative to the number of land use classes), we draw on recent literature that establishes both urban and agricultural nutrient sources.

Regarding fire, past studies indicate that wildfire effects scale with the proportion of a contributing area that is severely burned and thus some fires have little appreciable effect on water quality. It is not clear to what extent the catchments that are included in this study were impacted by high severity wildfire. There is a statement that > 1/3 of the area burned hot (lines 154-155). It’s unclear how that estimate corresponds to the specific study sites identified in Table 1. Please replot the map with watershed boundaries and burn severity. As mentioned above, provide additional detail regarding the forest, urban and agricultural areas.

High resolution burn severity data are available for the wildfires in our study. We now report the specific percentage of low, medium, and high severity wildfire in addition to the overall area burned in each watershed. This question of how much burned area is necessary before eliciting a hydrochemical response is of great interest, but with only 10 locations (some of which are nested), it is beyond the scope of this study to answer this question directly. Instead, we compare the effects of wildfire and various land uses on sediment, organic matter, and nutrient dynamics. However, the large sediment response and smaller but still significant nutrient response establishes that the wildfire scars were extensive and severe enough to change these parameters, despite being smaller than the direct human effects. We have revised the methods to add detail and the discussion to better contextualize these results.

The combined effects of the wildfire and storm are interesting, but it’s difficult to know how to interpret the findings from this combined disturbance. Would there have been no water quality effects without a severe storm? It seems like that the patterns would have been different for a less severe storm event. What is the threshold, for example of post-storm nutrient losses from the various agriculture or agricultural areas? If this was an unprecedented storm it is not surprising that the nutrient losses would exceed the conditions of stormwater collection and urban and agricultural water quality best management practices. Quite simply, the authors need to clarify if these findings are a unique response to the combined fire and storm and if so this qualification should be presented and discussed from the outset. The paper currently presents the findings as response to wildfire across land use/land cover types rather than a combined fire x storm event.

Throughout the methods and discussion, we now address the question of how representative this event is of past, current, and future disturbances. One of our general hypotheses is that as humans extend the wildfire season via climate change and ignition sources, more severe and extensive wildfires are likely to occur during shoulder seasons when high flow conditions can transport sediment and nutrients. The occurrence of early and late season megafires throughout the western US in the three years since our observations indicate that this pattern is becoming more dominant. That said, the rain event was not entirely unprecedented, and we now compare this event to other rain events in central Utah.

19 Specify the dates of the fire and storm in the abstract so readers will know how long it had been between the fire, flood, sampling, etc. – 

Added.

Line 165 Table 1

What is meant by herbaceous? Where are range or shrublands included? 

We apologize for the confusion. This is a standard category in the National Land Cover Database, but we now provide the official description of the class.

Specify the type of agriculture and especially whether this includes irrigated or fertilized row crops. 

Unfortunately, detailed agricultural data are not available in a spatially explicit format. We now provide a description of the types of agricultural uses within the watershed, though we cannot provide quantitative proportions.

Specify if the urban areas include suburban residential developments with irrigated and fertilized lawns. 

Added.

Line 175 Is grazing classified with agriculture? 

We do not include rangeland grazing as agricultural cover, and we have added a description of this.

Line 207 Where are all these analytes reported? 

We report all the analytes in the supplementary information and attached database. For the purposes of the study, we focus on the parameters that are most closely related to sediment, organic matter, and nutrients. We considered not mentioning the other analytes but wanted to alert readers to the existence of these data for potential future analysis. We now mention this justification in the text.

Line 371 Should this be Fig 8? 

This line refers to Fig. 8, which shows the inorganic and easily mineralizeable nitrogen and phosphorus. We have added a line of explanation to avoid confusion.

Discussion

377-378 It’s unclear how the hypothesized interaction reflects the direct human footprint? It seems that the stated response (higher nutrients/less bioavailable OM) relates to burning, but not human activity? Please revise to clarify the link. 

We have clarified this point in the discussion. The previous phrasing was confusing. We observed higher sediment loading from burned areas, but higher (or as high) nutrient loading from the areas with greater direct human influence. 

384-386 Not sure this sentence is needed as a segue.- 

We have removed the sentence. 

394 past tense? ‘amounted to’- 

Changed.

439-440 The loss of terrestrial plant matter would be a year/multi-year response. During this study you’re measuring the initial flush of charred, partially charred and unburned OM. This makes sense over a long-term, but in the immediate post-fire period, the export of- 

Thank you for this interesting insight. Other studies have observed an immediate decrease in DOM concentration and flux, though that could be due to a lack of hydrological connectivity immediately after the burn (i.e. the charred material was available but not transported to the surface water network). We have added some discussion around this.

448-450 Numerous studies have reported increased DOC and reactive C following both wildfire and lab heating experiments. 

We have revised this portion of the discussion and now more fully integrate the literature on pyrogenic DOM.

463-464 What experiment?- 

We now specify that this refers to the optical property measurements

465 “second factor” ? 

We have changed this to second

470 Yes, and scour during the flood would also have removed any existing algae/biofilms, so in stream autrophs would not have had much effect during the period of this study.

Thank you.

471 Are you referring to increased nutrient transfer from burned soils/uplands? 

Yes, because vegetation was eliminated, nutrient demand in upland and riparian soils would be decreased at the same time that transpiration would be low, leading to high nutrient export.

477 The post-fire increase in nutrients will contribute to and augment these concerns. Include it to the list here. 

Added 

479-510 The authors’ clear opinions make for interesting reading here.

Thank you.

Reference Format Please check and standardize.

Corrected throughout.

Additional References to consider

Cawley, K.M., A.K. Hohner, D.C. Podgorski, W.T. Cooper, J.A. Korak, and F.L. Rosario-Ortiz. 2017. Molecular and Spectroscopic Characterization of Water Extractable Organic Matter from Thermally Altered Soils Reveal Insight into Disinfection Byproduct Precursors. Environmental Science & Technology 51:771-779.

Chow, A.T., K.-P. Tsai, T.S. Fegel, D.N. Pierson, and C.C. Rhoades. 2019. Lasting Effects of Wildfire on Disinfection By-Product Formation in Forest Catchments. Journal of Environmental Quality 48:1826-1834.

Hohner, A.K., K. Cawley, J. Oropeza, R.S. Summers, and F.L. Rosario-Ortiz. 2016. Drinking water treatment response following a Colorado wildfire. Water Research 105:187-198.

Hohner, A.K., C.C. Rhoades, P. Wilkerson, and F.L. Rosario-Ortiz. 2019. Wildfires Alter Forest Watersheds and Threaten Drinking Water Quality. Accounts of Chemical Research 52:1234-1244.

Writer, J.H., A. Hohner, J. Oropeza, A. Schmidt, K. Cawley, and F.L. Rosario-Ortiz. 2014. Water treatment implications after the High Park Wildfire in Colorado. J. Am. Water Works Assoc 106:85-86.

We are grateful for these pertinent and interdisciplinary references, which we have integrated into the revision. More generally, thank you for the careful read and criticisms throughout the manuscript.

---

## [Editor Report · Decision Letter 1]

9 Sep 2021

Megafire affects stream sediment flux and dissolved organic matter reactivity, but land use dominates nutrient dynamics

PONE-D-21-06624R1

Dear Dr. Crandall,

We’re pleased to inform you that your manuscript has been judged scientifically suitable for publication and will be formally accepted for publication once it meets all outstanding technical requirements.

Kind regards,

Angelina Martínez-Yrízar, Ph.D.

Academic Editor

PLOS ONE

Additional Editor Comments (optional):

I have a few minor comments:

I am concerned with the use of the word “pristine”, when you are referring to a site “with minimum direct human influence” (L460) or with “less than 10% urban and agricultural land use” (L192).  I suggest you make this clear from the beginning of the ms. the first time you use this word. Are the “natural” sites the ones you are considering as “pristine”? 

Correct the number of this figure (L404), should be Fig. 8.

The use of the hyphen in the word “land-use” is not consistent throughout the text. Please correct.
---

## [Editor Report · Acceptance letter]

13 Sep 2021

PONE-D-21-06624R1 

Megafire affects stream sediment flux and dissolved organic matter reactivity, but land use dominates nutrient dynamics in semiarid watersheds 

Dear Dr. Crandall:

I'm pleased to inform you that your manuscript has been deemed suitable for publication in PLOS ONE. Congratulations! Your manuscript is now with our production department. 

Kind regards, 

on behalf of

Dr. Angelina Martínez-Yrízar 

Academic Editor

PLOS ONE